# Advances in Lipid-Based Nanoparticles for Cancer Chemoimmunotherapy

**DOI:** 10.3390/pharmaceutics13040520

**Published:** 2021-04-09

**Authors:** Tianqi Wang, Yusuke Suita, Saradha Miriyala, Jordan Dean, Nikos Tapinos, Jie Shen

**Affiliations:** 1Department of Biomedical and Pharmaceutical Sciences, University of Rhode Island, Kingston, RI 02881, USA; tianqi_wang@uri.edu (T.W.); jordan_dean@uri.edu (J.D.); 2Laboratory of Cancer Epigenetics and Plasticity, Brown University, Rhode Island Hospital, Providence, RI 02903, USA; yusuke_suita@brown.edu (Y.S.); saradha_miriyala@brown.edu (S.M.); 3Department of Neurosurgery, Warren Alpert Medical School, Brown University, Providence, RI 02903, USA; 4Department of Chemical Engineering, University of Rhode Island, Kingston, RI 02881, USA

**Keywords:** chemoimmunotherapy, lipid-based nanoparticles, liposomes, cancer therapy, immunotherapy

## Abstract

Nanomedicines have shown great potential in cancer therapy; in particular, the combination of chemotherapy and immunotherapy (namely chemoimmunotherapy) that is revolutionizing cancer treatment. Currently, most nanomedicines for chemoimmunotherapy are still in preclinical and clinical trials. Lipid-based nanoparticles, the most widely used nanomedicine platform in cancer therapy, is a promising delivery platform for chemoimmunotherapy. In this review, we introduce the commonly used immunotherapy agents and discuss the opportunities for chemoimmunotherapy mediated by lipid-based nanoparticles. We summarize the clinical trials involving lipid-based nanoparticles for chemoimmunotherapy. We also highlight different chemoimmunotherapy strategies based on lipid-based nanoparticles such as liposomes, nanodiscs, and lipid-based hybrid nanoparticles in preclinical research. Finally, we discuss the challenges that have hindered the clinical translation of lipid-based nanoparticles for chemoimmunotherapy, and their future perspectives.

## 1. Cancer Chemoimmunotherapy

Cancer chemoimmunotherapy is a treatment that utilizes the synergistic benefits of chemotherapy and immunotherapy. Chemotherapy typically involves the use of conventional cytotoxic drugs and/or novel molecularly targeted agents. On the other hand, immunotherapy, including the use of immune checkpoint inhibitors, adoptive cell therapy, cancer vaccines, and cytokines, is a fairly novel type of cancer therapy that uses the patient’s own immune system to attack cancer cells. We will introduce widely used cancer immunotherapy strategies and explain the advantages of cancer chemoimmunotherapy based on the understanding of combination mechanisms.

### 1.1. Cancer Immunotherapy

Since the first immunotherapy cytokine interferon-α (IFNα) was approved by the U.S. Food and Drug Administration (FDA) in 1986 for hairy cell leukemia, early-stage cancer immunotherapy from cytokines to interleukin-2 (IL-2) has shown limited therapeutic effect with high toxicity [1,2]. This situation was greatly improved when the pioneering checkpoint inhibitor ipilimumab, a monoclonal antibody (mAb) that targets CTLA4 (cytotoxic T-lymphocyte-associated protein 4), was approved for the treatment of advanced melanoma in 2011 by the FDA, which was featured by *Science* as the breakthrough of the year in 2013 [3,4]. Until now, cancer immunotherapy has been shown to be effective in treating certain cancers and has been approved by the FDA to treat melanoma, non-small cell lung cancer (NSCLC), kidney, bladder, head and neck, gastric, hepatocellular, and cervical cancer [5]. Over the past several years, cancer immunotherapy has been focused on immunosurveillance mechanisms, including release of tumor-associated antigens, tumor antigen-presenting cells (APCs), T-cell activation and trafficking, and the role of certain costimulatory factors (Figure 1) [6,7,8]. Based upon these mechanisms, cancer immunotherapy includes the following categories: immune checkpoint inhibitor therapy, adoptive cell therapy, vaccines, and cytokines [9].

#### 1.1.1. Immune Checkpoint Inhibitor (ICI) Therapy

ICI therapy is described as the use of therapeutic antibodies that interrupt the coinhibitory T-cell signaling pathways and unleash antitumor immune responses [10]. The development of ICIs is a revolutionary milestone in the field of immune oncology [11]. Ipilimumab, targeting CTLA-4, was the first ICI approved by the FDA for metastatic melanoma [3,12]. Following that, anti-programmed death (PD)-1 antibodies (e.g., pembrolizumab and nivolumab) and anti-programmed death ligand-1 (PD-L1) antibodies (e.g., atezolizumab and durvalumab) were developed and widely used in the treatment of several cancer types, including melanoma, NSCLC, renal cell carcinoma, and head and neck squamous cell carcinoma [13,14,15,16,17]. Building upon the recent success of ICIs, more than 3000 clinical trials using ICIs as either a single agent or in combination with chemotherapies are in progress for around 50 cancer types [11,18].

Although ICIs have shown success in cancer treatment, only a fraction of patients could benefit from these treatments because the antitumor immune response is modulated by several factors [10,19]. The ICIs showed higher responses in patients with certain biomarkers, resulting in a narrow therapeutic window. Combination strategies (e.g., using two ICIs or a combination of an ICI and chemotherapy), are thought to widen the therapeutic window of ICIs.

#### 1.1.2. Adaptive Cell Therapy (ACT)

ACT, including the use of tumor-infiltrating lymphocytes (TILs), engineered T-cell receptors (TCRs) and chimeric antigen receptors (CARs), is another attractive treatment modality in cancer immunotherapy. Compared with ICI therapy, ACT seems to be a more personalized treatment using autologous T lymphocytes of individual patients.

TILs extracted from fresh tumor samples or peripheral blood lymphocytes of patients, containing cluster of differentiation CD4+ and CD8+ T cells, were proven to mediate objective regression of cancer in patients with metastatic melanoma [20,21]. However, not all patients have the TILs that can recognize the tumor antigens. Researchers found that T cells could be collected from patients and engineered to express a TCR that could target a specific tumor antigen [22]. To generate TCRs, coculturing T cells with tumor APCs and genetic engineering was used to produce T cells with the desired TCRs [23]. Adoptive transfer of sorted New York esophageal squamous cell carcinoma-1 (NY-ESO-1) TCR T cells could specifically recognize tumor antigens and mount productive antitumor cell responses [24]. Both TILs and TCRs require antigen presentation via the major histocompatibility complex (MHC). In some cancer types, MHC expression is downregulated to escape from the immune system [25]. To solve this problem, CAR was developed. CD19, which is expressed on B-cell leukemias and lymphomas, was the initial target for CAR T cells [26,27,28,29]. In 2017, the first CAR T-cell therapy was approved by the FDA for the treatment of certain types of large B-cell lymphoma [30]. Compared with TILs and TCR, CAR T cells could retain their activity for a long time after one intravenous injection [31]. The clinical success of CAR T cells has encouraged the recent efforts to engineer other immune cells, such as natural killer (NK) cells. Recently, CAR NK cells have been explored in clinical trials for the treatment of several cancer types including B-cell lymphoma (NCT03692767), metastatic solid tumors (NCT03415100), ovarian cancer (NCT03692637), and prostate cancer (NCT03692663).

Overall, ACT is a more complex and expensive approach for cancer treatment than other types of immunotherapy. The “off-target” toxic effects caused by expression of antigens on normal cells has remained a challenge in ACT applications. Appropriate delivery strategies and more specific antigens are needed for the widespread applications of ACT.

#### 1.1.3. Cancer Vaccines

Vaccines have been proven to be effective in preventing diseases caused by viruses and bacteria. However, the development of cancer vaccines is more complicated and difficult compared with conventional vaccines for infectious diseases. Unlike viruses and bacteria, cancer cells can camouflage themselves as normal healthy cells. Furthermore, the tumor antigens between individual patients are quite different and unique [32]. These obstacles have hindered the development of cancer vaccines. To circumvent these issues, multiple cancer vaccines are currently being studied, including dendritic cell (DC)-based, nucleic acid-based, and neoantigen-based vaccines [33,34,35].

The DC vaccine sipuleucel-T (Provenge^®^) is the only cancer vaccine to date that has been approved by the FDA, in 2010, for prostate cancer treatment [36]. DC vaccines made from patients’ own stimulated DCs could prolong overall survival [37]. Nucleic acid vaccines, which deliver genes encoding tumor antigens by DNA or mRNA, is a promising strategy for harnessing the immune system against cancer [38]. Several clinical trials using plasmid DNA vaccines have demonstrated a good safety profile and the activation of a broad and specific immune response. However, therapeutic effects appeared to be modest due to the complicated immunosuppressive mechanisms and the barriers of nuclear delivery [38,39,40]. In particular, mRNA is easily degraded by nucleases and has poor storage stability [41]. Therefore, delivery platforms that increase intracellular uptake of vaccines to improve targeting effect as well as stability are important for the development of nucleic acid vaccines. Most recently, neoantigen vaccines have been investigated to overcome the obstacles in developing cancer vaccines, including lack of tumor specificity and poor immunogenicity [42]. The novel epitopes of self-antigens generated by the mutations of tumor cells are so-called neoantigens or neoepitopes [43]. The main advantage of neoantigens is that the tumor-specific antigen is only present in cancer cells, thus eliminating the “off-target” toxicity to nonmalignant tissues [44]. Furthermore, the neoantigens are de novo epitopes derived from somatic mutations, which could circumvent T-cell central tolerance to boost tumor-specific immune responses [45].

Numerous studies on different cancer vaccines are currently underway. In addition, the combination of cancer vaccines with other cancer therapy are in clinical trials with the goal of long-lasting, tumor-specific immunotherapy [46].

#### 1.1.4. Lymphocyte-Promoting Cytokines

Cytokines are molecules that elicit cellular functions by binding to the corresponding receptors on the surface of the cells, as exhibited by the etymology of the words (“cyto” meaning “cell” and “kinos” meaning “movement”). Cancer cells and immune cells secrete cytokines that bind to their corresponding receptors in order to communicate with each other. In cancer specifically, tumor cells use cytokines to instruct immune cells to create a tumor microenvironment (TME) that promotes the progression of the tumor. On the other hand, some cytokines have the opposite role—these cytokines instruct immune cells to attack cancer cells, and this immunogenic mechanism can be harnessed as a therapy. In particular, only some lymphocyte-promoting cytokines which are delivered systemically can be used for cancer immunotherapy. Until now, three cytokines (i.e., IL-2, IFN-α, and granulocyte-macrophage colony-stimulating factor (GM-CSF)), have been approved by the FDA for the treatment of several cancer types. For example, the IL-2 cytokine aldesleukin (Proleukin^®^) targeting the IL-2/IL-2R pathway was approved for subsets of patients with kidney cancer and melanoma [47,48]. IFN-α, including IFN-α-2a and IFN-α-2b (Intron A^®^ and Sylatron^®^), was approved for subsets of patients with leukemia, sarcoma, lymphoma, and melanoma [48,49,50,51]. In 2020, a new cytokine, GM-CSF, combined with naxitamab-gqgk (Danyelza), a targeted antibody against the GD-2 pathway, was approved for the treatment of relapsed/refractory neuroblastoma in the bone or bone marrow [52]. The clinical use of these cytokines marked a milestone in cancer immunotherapy. However, some problems still remain in cytokine-based immunotherapy. For example, high doses of IL-2 and IFN-α caused high toxicity, whereas low-doses of IL-2 had a low response rate [48,53,54,55]. Even so, cytokines such as IL-12, IL-15, and IL-21 combined with other approaches (such as ICI therapy and chemotherapy) are currently in clinical trials. These combination approaches present a more favorable opportunity to achieve an optimal cytokine balance [56].

### 1.2. Cancer Chemoimmunotherapy

Chemotherapy has served as the standard-of-care cancer treatment for a long time. It has been shown that the combination of chemotherapy and immunotherapy is a potential strategy in cancer therapy due to the synergistic effects on the TME. However, chemoimmunotherapy is not simply the result of combining chemotherapy and immunotherapy, since the mechanisms of these therapeutic modalities could influence each other. Most chemotherapeutic agents are cytotoxic agents with non-specific targets, which may not only cause the death of cancer cells but also influence the proliferation of most cell types in the human body [57]. The direct effects of chemotherapy on cancer cells (“on-target” effects) might elicit anticancer immune responses, whereas the “off-target” effects on the host immune cells might have implications for anticancer immunosurveillance [58]. The chemoimmunotherapy is either synergistic or complementary, depending on whether chemotherapeutic agents are immune-promoting or immune-suppressing. We summarize clinical chemoimmunotherapy strategies to illustrate the combination mechanisms in Table 1.

Immunogenic cell death (ICD), which is characterized by the secretion of damage-associated molecular patterns (DAMPs), is a widely known mechanism to stimulate the immune response through activation of APCs and consequent activation of a specific T-cell response [59,60]. Some cytotoxic chemotherapeutic agents, such as anthracyclines, cyclophosphamide, oxaliplatin (OXA), and bortezomib, can induce ICD effects by increasing the antigenicity of cancer cells [61,62,63,64]. Combination of ICD inducers with immune therapeutics might induce a synergistic effect to enhance immunotherapy. For example, doxorubicin (DOX) combined with the indoleamine 2,3-dioxygenase (IDO-1) inhibitor indoximod (IND) exhibited a synergistic antitumor response that was superior to a DOX-only treatment in 4T1 orthotopic tumor-bearing mice [65]. A combination of paclitaxel (PTX) and anti-PD-1 antibody activated the antitumor response by the ICD effect and suppressed the immune escape, resulting in a synergistic antitumor effect in a mouse melanoma model [66]. The apoptotic cells and cell fragments induced by the ICD effect of chemotherapeutic agents are also regarded as a potential source of cancer vaccine immunogen with personalization between individuals. Researchers utilized the concept of in situ vaccination to inject DOX intratumorally to generate immunogenic apoptotic fragments/cells and then combined them with a CpG oligodeoxynucleotides (CpG) adjuvant for in situ chemoimmunotherapy [67].

Multiple chemotherapeutic agents can interact with immune cell subsets directly to activate antitumor responses. Depletion of immunosuppressive cells is one of the positive effects of these chemotherapeutic agents. For example, low-dose 5-fluorouracil (5-FU) chemotherapy inhibited lung-accumulating myeloid-derived suppressor cells (MDSCs) in tumor-bearing mice [68]. Low-dose cyclophosphamide has been shown to deplete tumor-infiltrating regulatory T cells (T_reg_) cells in mice [69,70]. Some chemotherapeutic agents also modified the phenotype of immune cells to improve antitumor response. The repolarization of tumor-promoting M2-type tumor-associated macrophages (TAMs) into the tumor-suppressing M1 type has been observed in mice upon administration of PTX [71]. Combination of these chemotherapeutic agents with other immune activators seems to provide enhanced antitumor immune response in the TME.

In addition to the effects on cancer and immune cells directly, some conventional chemotherapeutic agents may cause some “off-target” side effects on the whole-body immune response involving multiple pathways, such as gastrointestinal toxicity affecting the gut microbiota and upregulation of systemic immunosuppressive factors [72,73]. Some novel chemotherapeutic agents such as molecularly targeted agents are also involved in immunosuppression caused by hypoxia after treatment [74,75]. Compared with chemotherapy alone, the combination of these chemotherapeutic agents with immune modulators, which reverses the immunosuppressive microenvironment and improves chemotherapeutic effect, is regarded as a complementary mechanism to enhance antitumor effect. For example, the combination of anti-PD-1 antibody, the CXCR-4 inhibitor AMD3100, and sorafenib showed a superior antitumor effect compared to anti-PD-1 antibody combined with sorafenib alone. This was because AMD3100 prevented the polarization toward an immunosuppressive microenvironment after the sorafenib treatment [74,76]. Finally, CpG oligodeoxynucleotides could reverse the immunosuppression caused by treatment with 5-FU in murine hepatoma to inhibit tumor growth [77].

In summary, combination of chemotherapeutic agents with immunotherapy can augment antitumor response in the TME, thus improving cancer treatment.

## 2. Immune Microenvironment in Cancer: The Glioblastoma Paradigm

Cancer is one of the leading causes of death in the United States (U.S.), fueling the urgency to develop new and efficient cancer treatments. Especially for cancers of the brain or nervous system, mortality rates have been increasing over the past few years (NCI Cancer Trends Progress Report). Of these diagnoses, glioblastoma (GBM), a very aggressive brain tumor, has one of the highest mortality rates. GBM has an average overall survival of 15 to 21 months after first diagnosis and a 5-year survival rate of less than 5%, which is the lowest among other central nervous system (CNS) tumors and second lowest out of all other types of cancer [78,79,80]. These numbers have slightly improved as existing therapies such as surgical removal, chemotherapy, and radiotherapy are used in conjunction with new therapeutic advances such as the use of oncolytic viruses, dendritic cell vaccines, tumor-treating fields, and immunotherapy [81,82,83,84,85]. However, these treatment options have failed to effectively treat GBM patients as they often result in eventual relapse or death. A facet of GBM that contributes to its dismal prognosis is the immunosuppressive nature of the GBM TME. Although functional lymphatic vessels in the brain allow the transport of immune cells from the cerebrospinal fluid, and by extension, the deep cervical lymph nodes, immune cells are unable to have an effect of the tumor because of the nature of the TME. The immunosuppressive effect of the TME is created and maintained using two mechanisms. In the first, the GBM immune environment promotes protumor activity by promoting regulatory T-cell [86] or MDSC [87] activity by inducing the recruitment of those tumor-suppressive cells to the tumor site and polarizing TAMs toward the immunosuppressive M2 phenotype [88]. Second, the GBM immune environment suppresses antitumor activity by inhibiting the activation and recruitment of cytotoxic T cells and preventing the migration of tumor-infiltrating T cells [89] and NK cells [90] to the tumor site, among other mechanisms.

Possible chemoimmunotherapeutic approaches to GBM must aim to modulate the immunosuppressive TME that allows GBM to progress. Specifically, these chemoimmunotherapeutic approaches must target cytokines, their binding receptors, and other associated cells that contribute to immunosuppression. Cytokines can be categorized into two broad groups based on their overall effect on the tumor and TME, and then further divided into subgroups based on the specific location and targets of these effects:(1)Protumor cytokines: Cytokines exhibit protumor effects through multiple possible mechanisms.Cytokines change the activity of protumor cells including regulatory T cells and MDSCs. IL12 can change the expression profile of proinflammatory cytokines by TAMs [91]. Chemokine (C-X-C motif) ligand 16 (CXCL16)/chemokine receptor 6 (CXCR6) signaling elicits anti-inflammatory effects in glioma by driving microglia polarization [92].Cytokines instigate the lymphatic migration of immune-tolerogenic APCs. Although DCs are responsible for immunogenicity via priming, they are also responsible for immunologic tolerance [93,94]. Particularly, DCs in the presence of tumor growth factor (TGF) beta-1 can transform cytotoxic T cells into T_reg_ cells [95].Cytokines released from tumor cells can recruit tumor-suppressive cells to the tumor site. IL33 produced by tumor cells can recruit TAM [96]. C-C motif chemokine ligand 2 (CCL2) produced by CXCR4+ tumor-associated microglia M1 can recruit the CC chemokine receptor 4 (CCR4)+-expressing T_reg_ cells and MDSCs to gliomas [97,98]. CCL2 is also known as the CCR4 ligand that is secreted from GBM and is responsible for T_reg_ accumulation in GBM [99].Cytokines can create tumor-immune tolerogenic response at the tumor-draining lymph nodes. Glioma cells secrete IDO, which prevents the activation of CD8/CD4+ cells by increasing T_reg_ activity [100,101]. Cytotoxic T-lymphocyte-associated protein 4 (CTLA4) on T cells suppresses the activation of CD4+ and CD8+ T cells [102].(2)Antitumor cytokines: Conversely, cytokines also exhibit antitumor activity.Cytokines induce immunologic response at the tumor site. IL12 activates T memory and effector cells [103]. IL27 increases NK cell activation and cytotoxicity against mammary tumor murine models [104]. Calreticulin, antigen expressed on the surface of cancer cells, can also activate DC phagocytosis. However, calreticulin normally resides inside of the cells, and needs to be translocated to the surface of the cancer cells to be detected by DCs [105].Cytokines recruit immunologic APCs. As stated previously, CCL2 can recruit tumor-associated microglia and MDSCs in the glioma TME, but also is known to induce the migration of antigen-presenting DCs in sarcoma and mammary carcinoma [106].Cytokines recruit cytotoxic T cells or/and NK cells. IL33 recruits CD4+ T helper cells and FOP3+ T_reg_ cells to the tumor site [96]. CXCL10 recruits NK cells to the tumor site [107].Cytokines can also induce an immunologic response at tumor-draining lymph nodes. The stimulator of interferon gene (STING) receptor and pathway on DCs is responsible for the production of type I and II IFNs, which activate T cells at tumor-draining lymph nodes [108].

In the case of GBM, immunotherapy has failed to show a significant effect [109]. It is believed that immunotherapy lacks efficacy in GBM due to the absence of an antitumor immunologic system, the presence of a protumor or immunosuppressive system, or both (Figure 2). A possible solution to augment the effects of immunotherapy in brain tumors may be the combinatorial use of chemo- and immunotherapy to: (1) Compensate for the absence of antitumor immune cells. Chemotherapy can be toxic to cancer cells and thereby create tumor-associated antigens that can be presented to naive antitumor T cells and transform the naive cells into CD8+ cytotoxic T cells; (2) Recruit immunologic cells. Chemotherapy can induce the secretion of promigratory factors from immune cells, and these promigratory factors can promote the recruitment of antitumor immune cells (i.e., cytotoxic T cells) to the tumor site; and (3) Reduce the activity of immunosuppressive cells. Chemoimmunotherapy can decrease the activity of immunosuppressive T_reg_ cells or MDSCs. Modulating this immune system can involve manipulating the activity of cytokines in the GBM TME. However, the GBM TME consists of a complex network of cytokine interactions. For example, CCL2 has been shown to induce the migration of DCs and to possess an immunogenic function and activate macrophages in conjunction with CXCL12 [106,110]. At the same time, CCL2 is known to be secreted from microglia and macrophages in the presence of CCL20 [110]. CCL2 shows not only an immunogenic effect but also an immunosuppressive effect. CCL2 promotes the recruitment of immunosuppressive T_reg_ and MDSCs, which can potentially suppress the antitumor immunologic activity of other immune cells [97]. Therefore, manipulating this network of cytokines in the GBM TME via chemoimmunotherapy requires technologies that comprehensively address the pre-existing intra- and intercellular interactions.

## 3. Lipid-Based Nanoparticles in Cancer Chemoimmunotherapy

Nanotechnology has attracted extensive attention in cancer therapy due to certain advantages [111]. For example, nanoparticles, including polymeric micelles, lipid-based nanoparticles, gold nanoparticles, and inorganic nanoparticles, are being widely used to deliver various therapeutic agents such as small molecules (either hydrophilic or hydrophobic), proteins, and genes for cancer therapy. These nanoparticles can deliver therapeutic agents to specific cells and organelles through either passive targeting mechanisms such as the enhanced permeability and retention (EPR) effect or active targeting mechanisms mediated by targeting ligands [112]. Due to the superior targeting effect of nanoparticle-based delivery systems to the tumor tissue and tumor cells, the distribution of therapeutics in the body is altered and consequently “off-target” side effects on the normal tissue can be decreased. These advantages will provide more opportunities for the application of nanoparticles in cancer chemoimmunotherapy to improve combination therapy with reduced side effects.

In particular, lipid-based nanoparticles possess attractive biological and versatile characteristics, including biocompatibility, biodegradability, and the ability to entrap both hydrophilic and hydrophobic therapeutics [113,114,115]. Moreover, surface properties (e.g., charge and decorating with targeting ligands) of lipid-based nanoparticles can be easily modified via varying lipid components or surface modification. Currently, lipid-based nanoparticles for cancer chemoimmunotherapy in preclinical studies include liposomes, nanodiscs, and lipid-based hybrid nanoparticles. Recent examples of lipid-based nanoparticles in cancer chemoimmunotherapy are listed in Table 2. Next, we will introduce different kinds of lipid-based nanoparticles and their advantages and disadvantages.

### 3.1. Liposomes

Liposomes, consisting primarily of phospholipids and cholesterol, are nanosized vesicles with demonstrated advantages in biocompatibility and enhanced targeted payload delivery with minimal toxicity. Amphiphilic phospholipids self-construct into a spherical lipid bilayer structure with their lipophilic tails, creating an environment for hydrophobic drugs to be encapsulated. On the other hand, hydrophilic heads of phospholipids assemble into an exterior surface and an aqueous core that can enclose hydrophilic agents. The encapsulation of various therapeutic agents into liposomes can occur through either charge–charge interactions or interaction with chemical linkers on the liposomal exterior. Importantly, the encapsulation of therapeutics within different liposomal compartments allows safe and targeted drug delivery, since the liposomes can protect the enclosed cargos from degradation by the immune system while carrying them across biological membranes that the free drugs (nonencapsulated in liposomes) are often incompatible with. Liposomes, enabling the drug delivery of both hydrophilic and lipophilic therapeutic agents while preserving their efficacy, are among the most successful nanotechnology-based drug products in cancer therapy. Since PEGylated liposomal DOX (Doxil^®^) became the first FDA approved nano-drug in 1995, more than six liposomal drugs have been approved by the FDA for use in cancer therapy. Building upon the success of liposomal drugs in chemotherapy, liposomes have been employed as one of the most attractive delivery vehicles in chemoimmunotherapy.

#### 3.1.1. Combinatorial Use of Liposomal Drugs and Immunotherapeutic Agents

Combining liposomal drugs with immunotherapeutic agents is a simple and acceptable method to accomplish chemoimmunotherapy. Some approved liposomal drugs, such as Doxil^®^, Taxol^®^, LipoTaxen^®^, and Onivyde^®^, have been used in chemoimmunotherapy. For example, a combination of Doxil^®^ with an immunostimulatory cytokine recombinant IL-18 (SB-485232) is currently in a clinical trial (NCT00659178). The phase I study showed that SB-485232 at a 3 mg/kg dose level in combination with Doxil^®^ was safe and biologically active for the treatment of ovarian cancer. This combination warranted further study in a phase II trial [144]. Another combination therapy based on Doxil^®^ is also in a phase I clinical trial (NCT00887796). The NY-ESO-1 vaccine, decitabine, and Doxil^®^ were administrated sequentially in 12 patients with relapsed epithelial ovarian cancer (EOC). The results showed that EOC cells treated with the combination of decitabine and Doxil^®^ promoted NY-ESO-1 antigen-restricted CTL recognition of EOC cell lines, and the effect of drug treatment on NY-ESO-1 expression was long-lasting [145]. Similarly, the improved cancer therapeutic effect was observed in a combination of Doxil^®^ and motolimod (a novel Toll-like receptor 8 agonist) in a phase I study (NCT01294293). The evidence of antitumor activity was observed at the 2.5 and 3.0 mg/m^2^ dose levels, and 3.0 mg/m^2^ was selected as the recommended phase II dose. Doxil^®^ plus motolimod costimulated antitumor innate and adaptive immunity mechanisms to achieve an enhanced antitumor effect [146].

There are other preclinical liposome-based chemoimmunotherapeutics, in which immunotherapeutic agents are encapsulated in liposomes to improve the delivery of these agents along with the use of liposomal drugs. Mahmoud et al. proposed a P5 peptide and Doxil^®^ combination strategy for HER2^+^ breast cancer treatment [117]. The results revealed that Doxil^®^ administrated before the immunotherapy could reduce the population and functions of MDSCs and enhance the subsequent immunotherapy, resulting in elevated CD4+ (*p* < 0.01) and CD8+ lymphocyte (*p* < 0.001) populations as well as IFN-γ production (*p* < 0.001). Compared to the free peptide and doxorubicin, Doxil^®^ plus liposomal P5 showed a decreased effect on MDSCs and tumor growth, which could be beneficial in breast cancer treatment. In another study, the combination of Doxil^®^ with liposomes containing another E75 immunogenic peptide (Lip-Pep) also exhibited enhanced antitumor effect in breast cancer [118]. It was shown that Doxil^®^ induced immune responses more effectively than the nonliposomal DOX. In the case of chemoimmunotherapy, the combination of Lip-Pep and Doxil^®^ was more effective and had a higher survival rate than the combination of Lip-Pep and nonliposomal DOX (*p* < 0.001) [118].

Combination of liposomal drugs with immunotherapeutic agents or liposomal immunotherapeutics is an efficient way to achieve enhanced chemoimmunotherapy with long-lasting response, high survival rate, and low toxicity. It is worth mentioning that sequential administration of chemo- and immunotherapeutics was used in some studies. Therefore, the proper dosing sequence of chemotherapy and immunotherapy is a key factor that needs to be taken into consideration for improved therapeutic effect.

#### 3.1.2. Immunoliposomes

Immunoliposomes, the structural combination of liposomes and antibodies, represent a new strategy for chemoimmunotherapy. Immunotherapeutic antibodies are decorated on the surface of liposomes and used with chemotherapeutic agents encapsulated in the liposomes. This two-in-one strategy may lead to enhanced chemoimmunotherapy with reduced toxicity. For example, Gu et al. developed a PD-L1 mAb modified pH-sensitive liposome for the combinatory use of docetaxel (DTX) and PD-L1 antibody for melanoma therapy. These immunoliposomes exhibited effective tumor inhibition and prolonged survival due to the synergistic effect of the activation of tumor-specific CD8+ T cells and highly selective tumor killing [130]. In recent years, novel immunoliposomes have been developed for chemoimmuntherapy. For example, a phosphatidyl choline reversed choline phosphate lipid (CP-Lip) was synthesized and modified with a PD-L1 antibody (CP-αPDL), as shown in Figure 3 [123]. DOX was loaded into the liposomes to achieve codelivery of DOX and αPDL (Dox@tCP-Lipos) to melanoma cells. Impressively, CP-Lip was able to insert into and interact strongly with the cell membrane, resulting in largely reduced fluidity and functionality of the membrane. As a result, the tumor was 100% suppressed after the treatment with Dox@tCP-Lipos.

Despite a large number of preclinical studies supporting the advantages of immunoliposomes, few immunoliposomes have eventually reached clinical trials. Some factors, such as lipid type, conjugation bond, as well as the combination ratio of chemotherapeutic and antibody, need to be optimized to enable the clinical translation of this new type of liposomes for cancer therapy.

#### 3.1.3. Codelivery of Chemo- and Immunotherapeutics in Liposomes

Two or more therapetics can be coencapsulated in one liposome and simultaneously delivered to the TME. Codelivery of chemo- and immunotherapeutics via the same liposome limits the delivery of both drugs into the same type of cells, which is particularly beneficial for immunotherapeutic agents with their target site in tumor cells. For example, the IDO inhibitor is a commonly used immunotherapeutic agent that targets tumor cells. Alkylated NLG919 (aNLG), an IDO1 inhibitor, and oxaliplatin-prodrug (Oxa(IV)) were coencapsulated in PEGylated liposomes as a bifunctional liposome (NLG/Oxa(IV)-Lip) for colorectal cancer therapy. NLG/Oxa(IV)-Lip not only released cytotoxic oxaliplatin inside the reductive cytosol to trigger immunogenic cell death (ICD) of cancer cells, but also efficiently retarded the degradation of tryptophan to immunosuppressive kynurenine via the NLG919-mediated inhibition of IDO1. NLG/Oxa(IV)-Lip showed synergistic antitumor effects in both subcutaneous and orthotopic CT26 tumor models [119]. In another study, DOX and indoximod (IDO inhibitor) were coencapsulated into bifunctional liposomes for the treatment of breast cancer. The chemoimmunotherapy mechanisms are illustrated in Figure 4 [65]. With the development of immunotherapy, more tumor-targeting immunotherapy will be investigated. It is anticipated that bifunctional liposomes will be widely used for the combination of these immunotherapeutic agents and chemotherapy.

### 3.2. Nanodiscs

Nanodiscs, composed of a lipid bilayer of phospholipids with the hydrophobic edge screened by two amphipathic proteins, also referred to as membrane scaffolding proteins (MSP), are a synthetic model membrane system. In some nanodiscs, the MSP is modified apolipoprotein A1 (apoA1), which is the main constituent in high-density lipoproteins (HDL). The structure of nanodisc is similar to discoidal HDL, which mimics a more native environment than liposomes and micelles. This biomimicking delivery system seems to be more effective in immunotherapy. Schwendeman’s group have done a lot of work on nanodisc-based chemoimmunotherapy. Initially, they developed an HDL-mimicking nanodisc coupled with a neoantigen (Ag peptide) and adjuvant (CpG) to draining lymph nodes for vaccination. The nanodisc elicited up to 47-fold greater frequencies of neoantigen-specific CTLs than soluble vaccines and 31-fold greater than the adjuvant (i.e., CpG in Montanide) used in clinical trials. These findings supported a new powerful approach for cancer immunotherapy [147]. The same group also employed nanodiscs for chemoimmunotherapy. They found that chemoimmunotherapy based on the combination of DOX-carrying nanodiscs and anti-PD1 induced complete regression of established CT26 and MC38 colon carcinoma tumors in 80–88% of animals and protected survivors against tumor recurrence [134]. Recently, they developed synthetic high-density lipoprotein (sHDL) nanodiscs loaded with CpG together with DTX for the chemoimmunotherapy of GBM (Figure 5) [135]. DTX-sHDL-CpG nanodiscs exhibited improved therapeutic effect with no overt “off-target” side effects. Furthermore, the combination of DTX-sHDL-CpG treatment with radiation (IR) resulted in tumor regression and long-term survival in 80% of GBM-bearing animals. In another study, Han et al. developed a lipophilic AS1411 aptamer-immunoadjuvant CpG fused sequences (Apt-CpG-DSPE)-modified HDLs for coloading of CpG and DOX (imHDL/Apt-CpG-Dox) for the treatment of lung cancer [137]. The imHDL/Apt-CpG-Dox was endocytosed into tumor cells as mediated by the recognition of AS1411 and nucleolin (sequential module II), translocating DOX to the nucleus and enabling tumor ablation and antigen release. The liberated CpG motif further evoked antigen recognition, induced vast secretion of proinflammatory cytokines, and potentiated host antitumor immunity. Although nanodiscs have demonstrated a huge potential for chemoimmunotherapy, there are some issues (such as the contact angle between nanodisc and immune cells and targeting specificity) that need to be solved.

### 3.3. Lipid-Based Hybrid Nanoparticles

Lipid-based hybrid nanoparticles with versatile structures (e.g., core–shell-structured hybrid nanoparticles with either a lipid core or shell) are attractive for chemoimmunotherapy. Some inorganic nanoparticles with a lipid shell have been developed for efficient therapeutic loading. For example, Kong et al. developed lipid-coated biodegradable hollow mesoporous silica nanoparticles (dHMLB) with coencapsulation of all-trans retinoic acid (ATRA), DOX, and IL-2 (A/D/I-dHMLB) for chemoimmunotherapy [139]. A/D/I-dHMLB exhibited a much higher tumor inhibitory rate (84.8 ± 13.0%) compared to that of any two-agent coloaded nanoparticles and a mixture solution of three agents. Similarly, Nano-Folox-encapsulated folinic acid (FnA) and oxaliplatin (OXA) were combined with 5-FU and anti-PD-L1 monoclonal antibody, resulting in decreased liver metastases in mice [140]. Nano-Folox was formed by a nanoprecipitate (C_26_H_35_N_9_O_7_Pt) core and an aminoethyl anisamide-targeted PEGylated lipid shell as shown in Figure 6. Compared to FOLFOX (the standard treatment of colorectal cancer), the significantly stronger chemoimmunotherapeutic responses were achieved by the combination of Nano-Folox and 5-FU without showing toxicity.

A lipid core of hybrid nanoparticles has also been used for loading drugs for chemoimmunotherapy. Zhang et al. developed twin-like core–shell nanoparticles (TCNs) for synchronous biodistribution and targeted delivery of sorafenib (SF) and IMD-0354 (an TAM repolarization agent) to cancer cells and TAMs to enhance tumor-localized chemoimmunotherapy, respectively [141]. The cationic lipid-based nanoparticles served as the core for loading different therapeutics, and they were modified using the same shell (O-carboxymethyl-chitosan) to achieve synchronous biodistribution (Figure 7). The results of in vivo antitumor effect and phenotype analysis of TAM in tumor tissues demonstrated that TCNs exhibited a superior synergistic antitumor effect and polarization ability of M2-type TAMs compared with the SF-only treatment in Hepa1-6 tumor-bearing mice.

In addition to core–shell-structured hybrid nanoparticles, other lipid composite nanoparticles have also been investigated for chemimmunotherapy. Huang et al. developed cationic lipid-assisted nanoparticles (CLANs) for codelivery of IDO1 siRNA and OXA for the treatment of orthotopic pancreatic cancer [142]. The CLANs were formed using a double emulsion solvent evaporation technique [148]. siRNA was successfully delivered to both tumor-draining lymph nodes and tumor tissues in the presence of cationic lipids. The contemporaneous administration of OXA and CLANs IDO1 could achieve synergetic antitumor effects via promoting DC maturation, increasing tumor-infiltrating T lymphocytes as well as decreasing the number of regulatory T cells in a subcutaneous colorectal tumor model. Recently, a novel hybrid nanoparticle was designed and used for the codelivery of GM-CSF and DTX in metastatic peritoneal carcinoma (mPC) therapy [143]. This hybrid nanoparticle was formed by genetically engineered exosomes and thermosensitive liposomes (gETL NPs), as shown in Figure 8. The gETL NPs administrated via intravenous injection could efficiently penetrate into mPC tumors and release payloads under the hyperthermia conditions of hyperthermic intraperitoneal chemotherapy (HIPEC).

The lipid-based hybrid nanoparticles preserve the advantages of lipid nanoparticles yet provide a more flexible structure, which could be advantageous in chemoimmunotherapy. However, the complicated structure and fabrication procedures will impede the clinical translation of these hybrid nanoparticles.

## 4. Challenges and Future Perspectives

Chemoimmunotherapy has been increasingly important in cancer therapy due to advantages such as high immune responsiveness, widened applications in different cancer types, as well as long-term therapeutic effect. Nanotechnology-based delivery systems, in particular lipid-based nanoparticles, have been used to achieve exciting and encouraging chemoimmunotherapy outcomes in preclinical research, due to their ability to reduce “off-target” side effects and deliver therapeutics (such as siRNA and proteins) to specific target sites while preserving their biological activity. However, the clinical translation of lipid-based nanoparticles for chemoimmunotherapy is still limited because some challenging issues are yet to be resolved. First of all, the design of lipid-based nanoparticles needs to be simplified in order to enable large-scale production. Some multifunctional lipid-based nanoparticles contain novel materials. The safety of these novel materials is yet to be studied. Secondly, the combination therapy requires a proper ratio and synergistic index of chemo- and immunotherapeutics, which need to be well controlled when using codelivery systems. Since some chemotherapeutic and immunotherapeutic agents have a narrow therapeutic window for cancer combination therapy, it is important to control the release amount and ratio of two types of therapeutic agents in the TME to achieve the desired synergistic effect. For example, DOX can cause an ICD effect at a lower concentration. When combining DOX with other immunotherapeutic agents based on the ICD effect, a low dose of DOX should be considered. Thirdly, some novel synthetic lipid-based nanoparticles may be immunogenic, thus resulting in side effects or immune response in the TME. Lastly, the mechanism of some chemoimmunotherapy strategies is still unknown. The delivery and dosing sequence of the combinatorial agents is another important factor that needs to be taken into consideration for improved anticancer effect. Typically, immunotherapeutic agents needed to be dosed two or more times with a long time interval to stimulate the immune response. This is not in accordance with the dosing frequency of some chemotherapies, which require the administration of chemotherapeutic agents frequently. Therefore, designing a proper delivery system which can codeliver combinational agents with different release behaviors is important. The delivery sequence is another factor to be considered in combination therapy. In some cases, chemotherapeutic agents need to be delivered prior to immunotherapy since the chemotherapy may trigger certain immune responses. On the other hand, if immunotherapy is needed to stimulate some immune response prior to chemotherapy, a codelivery system with delayed release of chemotherapeutic agents may be desirable. Combinations of immunotherapy with multiple therapeutic strategies (e.g., the combination of immunotherapy with radiotherapy or photothermal therapy as well as diagnosis) are currently being investigated in preclinical studies and have shown synergistic effects in cancer treatment. Lipid-based nanoparticles have been used in these novel combination strategies. Building upon the recent success of RNA vaccine therapies, it is expected that lipid-based nanoparticles could be an efficient delivery candidate for RNA drugs in cancer therapy. Lipid-based nanoparticles could enable improved synergistic effects of combination therapies in cancer treatment.

## Figures and Tables

**Figure 1 pharmaceutics-13-00520-f001:**
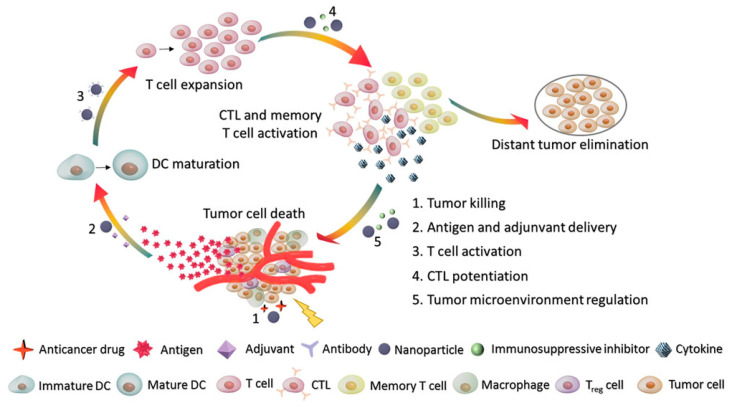
Cancer-immunity microenvironment affecting responsiveness to immunotherapy Adapted with permission from [7], Small 2019. CTL, cytotoxic T lymphocyte; DC, dendritic cell; T_reg_, regulatory T cell.

**Figure 2 pharmaceutics-13-00520-f002:**
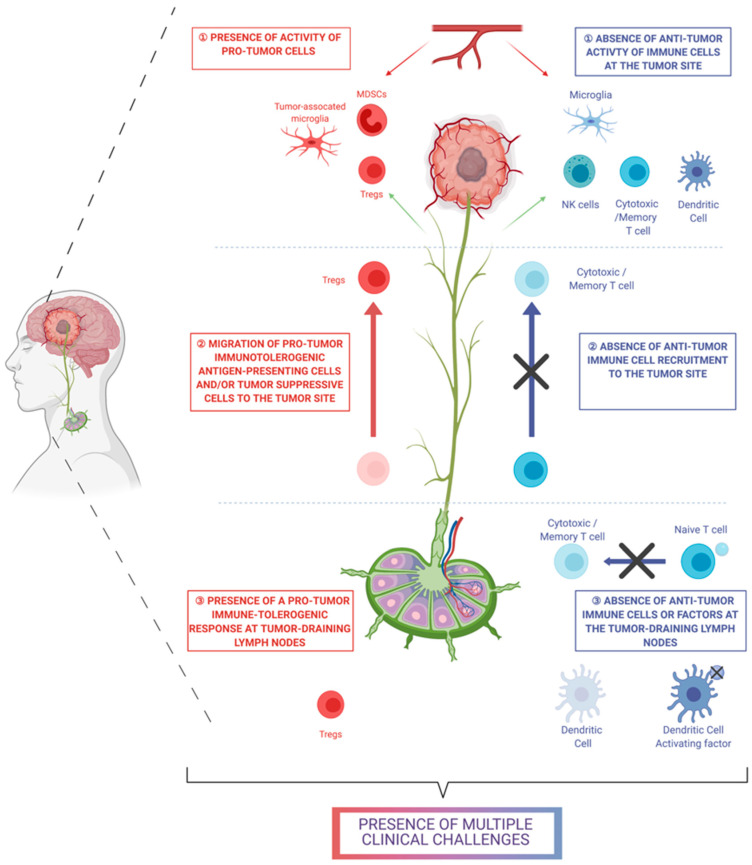
List of clinical challenges that need to be addressed by chemoimmunotherapy for glioblastoma (created with BioRender.com)**.**

**Figure 3 pharmaceutics-13-00520-f003:**
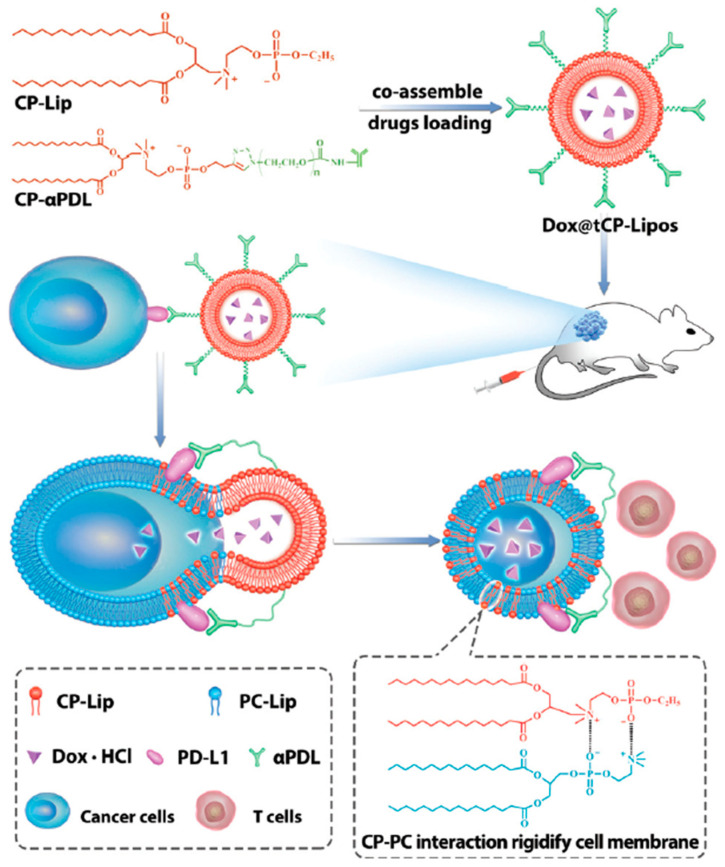
Schematic showing the chemical structure of phosphatidyl choline reversed choline phosphate lipid (CP-Lip) and the endocytosis mechanisms of liposomes codelivery of DOX and αPDL Dox@tCP-Lipos. Reproduced from [123], Chemical Communications, 2021.PC-Lip, phosphatidyl choline lipid; CP-Lip, choline phosphate lipid; DOX·HCl, doxorubicin hydrochloride; PD-L1, Programmed death-ligand 1; αPDL, PD-L1 antibody.

**Figure 4 pharmaceutics-13-00520-f004:**
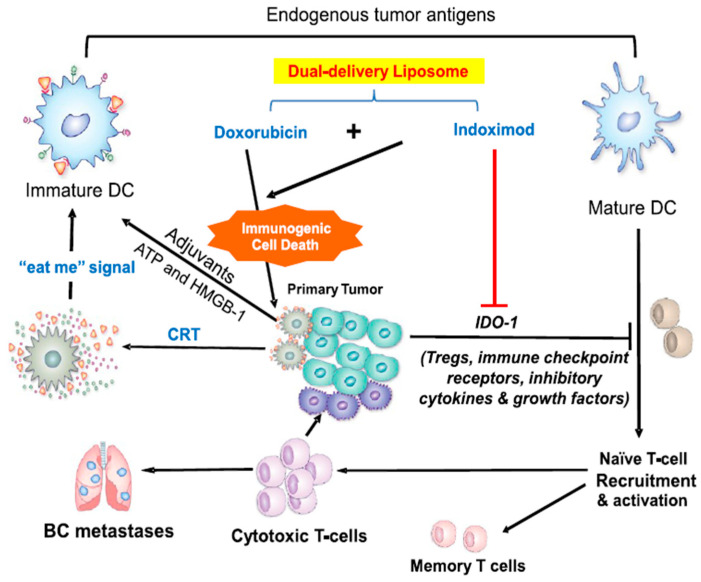
Schematic to explain breast cancer immunotherapy by codelivery of an immunogenic cell death stimulus plus an inhibitor of the IDO-1 pathway. Adapted with permission from [65], ACS Nano 2018. https://pubs.acs.org/doi/full/10.1021/acsnano.8b05189 22 February 2021. CRT, calreticulin; BC, breast cancer; HMGB-1, high mobility group box 1 protein.

**Figure 5 pharmaceutics-13-00520-f005:**
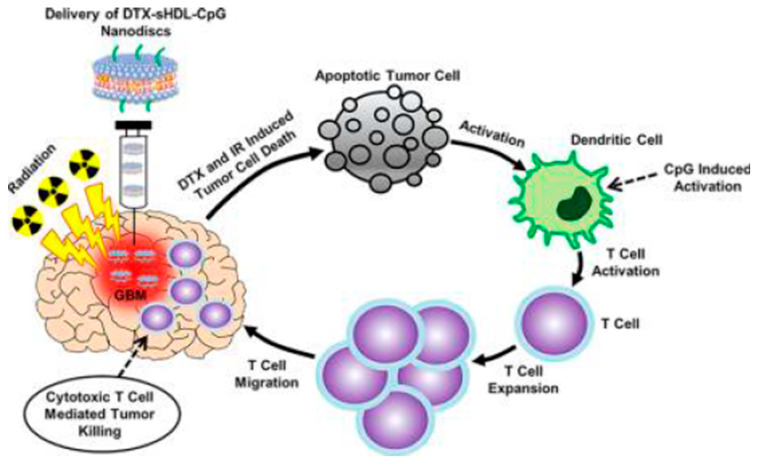
Immune-mediated antiglioma mechanism of docetaxel-loaded CpG-sHDL nanodiscs. Reproduced with permission from [135], ACS Nano 2018. CpG, CpG oligodeoxynucleotides; sHDL, synthetic high-density lipoprotein.

**Figure 6 pharmaceutics-13-00520-f006:**
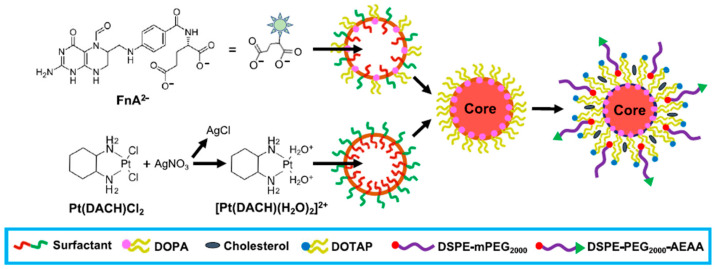
Schematic of Nano-Folox formulated in microemulsions using a nanoprecipitation technique. Reproduced with permission from [140], ACS Nano 2020. DOPA, 1,2-dioleoyl-sn-glycero-3-phosphate; DOTAP, 1,2-dioleoyl-3-trimethylammonium propane; DSPE-mPEG_2000_, 1,2-distearoyl-sn-glycero-3-phosphoethanolamine- polyethyleneglycol-2000; AEAA, aminoethyl anisamide.

**Figure 7 pharmaceutics-13-00520-f007:**
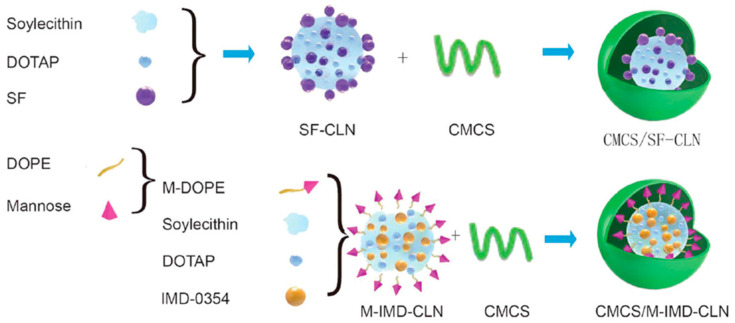
Schematic of twin-like core–shell nanoparticles (CMCS/SF-CLN + CMCS/M-IMD-CLN) for synchronous biodistribution and cell-specific targeted delivery to enhance chemoimmunotherapy. Adapted from [141]. Nanoscale 2019. SF, sorafenib; DOPE, 1,2-dioleoyl-sn-glycero-3-phosphoethanolamine; M-DOPE, mannose-DOPE; CMCS, carboxymethyl chitosan; CLN, cationic lipid-based nanoparticles; IMD, IMD0354, a selective IKKβ inhibitor.

**Figure 8 pharmaceutics-13-00520-f008:**
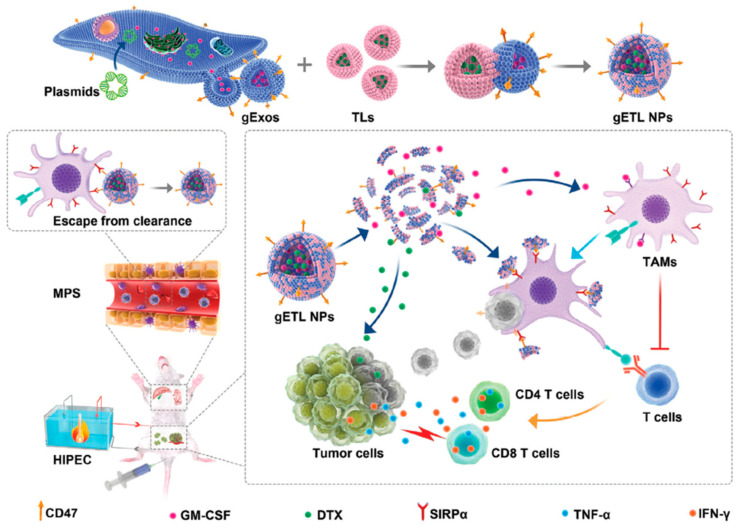
Schematic of the synthesis and application of genetically engineered exosomes and thermosensitive liposomes for the treatment of metastatic peritoneal carcinoma. Adapted from [143], Advanced Science, 2020. gETL NPs, genetically engineered exosomes-thermosensitive liposomes hybrid nanoparticles; HIPEC, hyperthermic intraperitoneal chemotherapy; TAMs, tumor-associated macrophages; gExos, gExos, genetically engineered exosomes; TLs, thermosensitive liposomes; MPS, mononuclear phagocytic system.

**Table 1 pharmaceutics-13-00520-t001:** Representative clinical trials of chemoimmunotherapy.

Chemotherapeutic Agents	Immunotherapeutic Agents	Cancer Type	Phase	Identifier
Cisplatin (CP) + docetaxel (DTX)	Dendritic cells	Head and neck cancer	I	NCT01149902
Irinotecan + temozolomide	GM-CSF	Neuroblastoma	I	NCT03189706
CP	Pembrolizumab	Ovarian cancer recurrent	I/II	NCT03734692
Fulvestrant + DTX	Trastuzumab, pertuzumab	HER2-positive breast cancer, ER-positive breast cancer	I/II	NCT02345772
CP, doxorubicin (DOX), etoposide	Epratuzumab	Recurrent childhood acute lymphoblastic leukemia	I/II	NCT00098839
Carboplatin, DOX	Tocilizumab + IFN-α-2b	Recurrent ovarian cancer	I/II	NCT01637532
DTX	Cemiplimab	Prostate cancer metastatic	II	NCT03951831
Carboplatin + etoposide + lamivudine	Atezolizumab	Extensive-stage lung small cell carcinoma	II	NCT04696575
CHOP (CP + DOX + vincristine + prednisone)	Alemtuzumab	Lymphoma, T cell, peripheral	II	NCT01679860
CHOP	Rituximab	Lymphoma, large B cell	II	NCT03536039
Carboplatin	GM-CSF, rIFN-γ	Ovarian cancer, fallopian tube cancer, peritoneal cancer	II	NCT00501644
Cisplatin, carboplatin, etoposide	Durvalumab, ceralasertib	Extensive-stage small cell lung cancer	II	NCT04699838
Cisplatin + gemcitabine	Atezolizumab	Bladder cancer	II	NCT04630730
Bendamustine + mitoxantrone	Rituximab	Follicular lymphoma	II	NCT01523860
Pemetrexed + carboplatin	Durvalumab	Non-small cell lung cancer, metastatic	II	NCT04163432
Bendamustine	Obinutuzumab	Mantle cell lymphoma, non-Hodgkin lymphoma	II	NCT03311126
Fludarabine + CP	Ofatumumab	B-cell lymphoid leukemia	II	NCT01762202
Cabazitaxel + carboplatin	Nivolumab + ipilimumab	Metastatic prostate neuroendocrine carcinoma, metastatic prostate cancer	II	NCT04709276
Fludarabine + CP	Ofatumumab	Small lymphocytic Lymphoma, chronic lymphocytic leukemia	II	NCT01145209
5-Fluorouracil	IFN	Colon cancer stage III	III	NCT00309530
Fludarabine + CP	Rituximab	Chronic lymphocytic leukemia	III	NCT03836261
DHAP (dexamethasone + cytarabine + CP)	Ofatumumab, rituximab	Relapsed or refractory diffuse large B cell lymphoma	III	NCT01014208
Venetoclax, ibrutinib	Rituximab, obinutuzumab	Chronic lymphocytic leukemia	III	NCT02950051
Carboplatin + paclitaxel	Nivolumab	Non-small cell lung cancer	III	NCT04564157

**Table 2 pharmaceutics-13-00520-t002:** Preclinical studies of lipid-based nanoparticles in cancer chemoimmunotherapy.

Formulation	Chemotherapy	Immunotherapy	Cancer Type	Administration Route	Ref
***Liposomes***					
PEGylated liposomes	Doxorubicin	Alendronate	Breast cancer	i.v.	[116]
	Doxorubicin	P5 peptide	Breast cancer	i.v.	[117]
	Doxorubicin	E75 immunogenic peptide	Breast cancer	i.v.	[118]
	Oxaliplatin	NLG919	Colorectal cancer	i.v.	[119]
PEGylated cleavable lipopeptide (PCL)-modified liposomes	Oxaliplatin	TLR7 agonist	Colorectal cancer, melanoma	i.v.	[120]
Phospholipid-conjugated prodrug liposomes	Doxorubicin	Indoximod	Breast cancer	i.v.	[65]
Anionic liposomes	Paclitaxel	Adenovirus encoding for murine interleukin-12	Melanoma	Intra-tumoral	[121]
Charge-reversal cell penetrating peptide-modified liposomes	Paclitaxel	PD-L1 antibody	Melanoma	i.v.	[122]
Choline phosphate lipid-based liposomes	Doxorubicin	PD-L1 antibody	Melanoma	i.v.	[123]
Heparin-coated pH-sensitive liposomes	Doxorubicin	Epacadostat	Metastasis melanoma	i.v.	[124]
pH-responsive liposomes	Mitoxantrone	Indoximod	Breast cancer and renal cancer	i.v.	[125]
Enzyme/pH dual-sensitive micelle-liposomes	Paclitaxel	HY19991	Metastatic breast cancer	i.v.	[126]
Immunoliposomes	Irinotecan	JQ1	Colorectal cancer	i.v.	[127]
	Bufalin	anti-CD40 antibody	Melanoma	i.v.	[128]
	Docetaxel	Trastuzumab	HER2-positive breast cancers	N/A	[129]
	Docetaxel	PD-L1 monoclonal antibodies	Melanoma	i.v.	[130]
	miR-130a+ Oxaliplatin	PD-L1 monoclonal antibody	PD-L1-positive gastric cancers	i.v.	[131]
Temperature-sensitive liposomes	Doxorubicin	B16-OVA/CpG	Breast cancer	s.c. and i.p.	[132]
Low-temperature-sensitive liposomes	Doxorubicin	High-intensity focused ultrasound	Colorectal cancer	i.v.	[133]
***Nanodiscs***					
HDL-Nanodisc	Doxorubicin	αPD-1	Colorectal cancer	i.v.	[134]
	Docetaxel	CpG	Glioblastoma multiforme	Intra-tumoral	[135]
	Docetaxel	Cholesterol modified CpG	Colon carcinomas	Intra-tumoral	[136]
	Doxorubicin	CpG	Lung cancer	i.v.	[137]
***Lipid-based hybrid nanoparticles***					
Lipid-coated calcium nanoparticles	Zoledronate	Zoledronate	Lung cancer	i.v.	[138]
Liposome-coated mesoporous silica nanoparticles	All-trans retinoic acid + doxorubicin	IL-2	Melanoma	i.v.	[139]
Nano-Folox	Folinic acid + 5-FU + oxaliplatin	anti-PD-L1 monoclonal antibody	Colorectal cancer	i.v.	[140]
Twin-like core–shell nanoparticles	Sorafenib	IMD0354	Hepatocellular carcinoma	i.v.	[141]
Cationic lipid-assisted nanoparticles	Oxaliplatin	IDO1-siRNA	Colorectal cancer	i.v.	[142]
Thermo-sensitive exosome-liposome hybrid nanoparticles	Docetaxel	GM-CSF	Metastatic peritoneal carcinoma	i.v.	[143]

i.v., intravenous injection; s.c., subcutaneous injection; i.p., intraperitoneal injection; 5-FU, Fluorouracil; siRNA, small interfering RNA.

## Data Availability

Not applicable.

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
