# Peer review of "Advances in Lipid-Based Nanoparticles for Cancer Chemoimmunotherapy"

_pharmaceutics, 2021, doi:10.3390/pharmaceutics13040520_

Round 1

Reviewer 1 Report

Comments 1. Can author give more highlight on Cancer vaccine. It could be better if he can summarize the data in Table also. 2. What is the role of lipid in cancer treatment. add in one para. 3. Mention the list of lipids reported for anticancer property with their concentration reported. 4. The lipids NPs having many drawbacks as delivery. How the author overcome for cancer delivery. 5. Author have written in manuscript as lipid coated NPs.....Write the name of lipid. 6. Can author add the combination therapy for bioactive with synthetic drug as lipid based delivery for cancer therapy. 7. Cite the following a. Molecular pharmaceutics 13 (11), 3773-3782. b. International Journal of Biological Macromolecules 116, 1260-1267. c. Journal of Drug Delivery Science and Technology, 102198

Reviewer 2 Report

I read with great interest the review by Wang and co-authors describing the use of lipid-based nanoparticles in cancer therapy. The review is well written and organized, describing the most remarkable nanoparticles for cancer therapy in preclinical development.  References are up-to-date and a really interesting introduction is given to the main aspect of cancer immunotherapy. The different types of lipid-nanoparticles are really well described and properly grouped. The review is overall attractive and interesting to the reader. In my opinion there is only a minor change to introduce to further improve the quality of the review making it acceptable for publication:

1) The authors should improve chapter 3 more widely describing the advantages given by nanopaticles in cancer therapy. The main concept are described but should be widely detailed.
